# Natural Polyphenols in Metabolic Syndrome: Protective Mechanisms and Clinical Applications

**DOI:** 10.3390/ijms22116110

**Published:** 2021-06-06

**Authors:** Shiyao Zhang, Mengyi Xu, Wenxiang Zhang, Chang Liu, Siyu Chen

**Affiliations:** State Key Laboratory of Natural Medicines and School of Life Science and Technology, China Pharmaceutical University, Nanjing 211198, China; 3119030138@stu.cpu.edu.cn (S.Z.); 1821030605@stu.cpu.edu.cn (M.X.); wenxiangzhang@cpu.edu.cn (W.Z.); changliu@cpu.edu.cn (C.L.)

**Keywords:** metabolic syndrome, natural polyphenol, protective effect, application

## Abstract

Metabolic syndrome (MetS) is a chronic disease, including abdominal obesity, dyslipidemia, hyperglycemia, and hypertension. It should be noted that the occurrence of MetS is closely related to oxidative stress-induced mitochondrial dysfunction, ectopic fat accumulation, and the impairment of the antioxidant system, which in turn further aggravates the intracellular oxidative imbalance and inflammatory response. As enriched anti-inflammatory and antioxidant components in plants, natural polyphenols exhibit beneficial effects, including improving liver fat accumulation and dyslipidemia, reducing blood pressure. Hence, they are expected to be useful in the prevention and management of MetS. At present, epidemiological studies indicate a negative correlation between polyphenol intake and MetS incidence. In this review, we summarized and discussed the most promising natural polyphenols (including flavonoid and non-flavonoid drugs) in the precaution and treatment of MetS, including their anti-inflammatory and antioxidant properties, as well as their regulatory functions involved in glycolipid homeostasis.

## 1. Introduction

The metabolic syndrome (MetS) is a cluster of coexisting metabolic risk factors [1]. According to the 2009 consensus between American several major organizations, patient was diagnosed with MetS if they were suffered from abdominal obesity, elevated blood pressure, elevated triglyceride (TG) level, reduced high-density lipoprotein (HDL)-cholesterol (HDL-C) level, and elevated fasting glucose [1]. MetS is primarily characterized by central adiposity, which is closed to insulin resistance and a facilitator to the development of other metabolic risk factors [1,2]. The prevalence of MetS on the basis of global epidemiological studies is between 20% and 45% [3], and is expected to increase to approximately 53% by 2035 [4]. Additionally, the development of MetS increases the risk factors of type 2 diabetes (T2D) and cardiovascular diseases (CVDs) [5,6,7], leading to an enormous pressure on the healthcare and social economy of the whole society [8,9]. Therefore, it is an urgent and critical challenge for biologists to determine active pharmaceutical ingredients or pro-drugs that improve MetS and its complications [10].

Nonalcoholic fatty liver disease (NAFLD) is the spectrum of fatty liver disorders independent of the abuse of the alcohol, the infection of virus, autoimmune, drug-induced, and genetic etiologies [11]. This disease has recently been renamed metabolic-associated fatty liver disease [12]. NAFLD is generally considered to be a hepatic manifestation of the MetS [13]. However, the increasing evidence suggests that NAFLD has a mutual and bi-directional association with the MetS [11,13]. In detail, NAFLD precedes and appears to contribute to the development of the MetS, and the stage of liver fibrosis has been a strong determinant of such a metabolic risk [13]. In addition, with the progression of NAFLD, there is a possibility of transition to nonalcoholic steatohepatitis (NASH) and hepatocellular carcinoma (HCC) in the future [14]. As the worst outcome of NAFLD, HCC is strongly associated with diabetes, suggesting that NAFLD is also the precursor of T2D [15]. Consequently, targeting NAFLD has the potential to be a therapeutic strategy for MetS.

MetS is also a pathological condition, characterized by chronic, systemic, and low-level inflammation and oxidative imbalance [16,17]. By this, antioxidant and anti-inflammatory compounds might have beneficial effects on the onset and progression of MetS [18]. Natural polyphenols are a class of biomolecules widely found in plants, with more than 8000 species, most of which exist in cocoa beans, nuts, soybeans, olives, sesame seeds, tea, red wine, vegetables, and fruits [19]. Their structure is mainly characterized by containing one or more phenolic groups, and can be simply divided into flavonoids and non-flavonoids [20]. Natural polyphenols generally have antioxidant and anti-inflammatory effects, both in vivo and in vitro [21,22]. In addition, most of them also strengthen blood vessel walls, promote gastrointestinal digestion, lower blood lipids, prevent atherosclerosis, lower blood pressure, enhance human immunity, and inhibit the growth of bacteria and cancer cells [23]. At present, some epidemiological studies have found a negative correlation between polyphenol intake and the incidence of NAFLD and MetS [18,24]. In this context, we summarize some natural polyphenols (including flavonoids and non-flavonoids) that have been reported to be involved in the prevention and management of MetS and its components, and review the possible mechanisms and clinical applications of these natural compounds (Figure 1 and Table 1).

## 2. Flavonoids

Flavonoids originally referred to a class of compounds whose parent nucleus is 2-phenylchromone [68]. However, they are now generally a series of compounds formed by the interconnection of two benzene rings (A-ring and B-ring) through a central three-carbon chain, namely, a class of compounds with the structure of C6–C3–C6 [69]. The parent nucleus of flavonoids often contains hydroxyl, methoxy, alkoxyl, isopentenoxyl, and other auxochrome groups [70]. Because of the existence of these chromophores, these compounds are usually yellow in appearance, hence the name flavonoids [71].

Flavonoids are widely distributed in fruits, vegetables, tea, red wine, grains, flowers, and so on [72]. Most flavonoids couple with saccharides to form glycosides in plants, and a few exist in free form [73]. In addition, they can be further classified into the following categories according to the degree of oxidation of the three-carbon bond structure and the location of the B-ring: flavones, flavonols, dihydroflavones, dihydroflavonols, isoflavones, dihydroisoflavones, chalcones, aurones, flavanes, anthocyanidins, biflavones, etc. [74]. Among them, flavones and flavonols are the largest subgroups of polyphenols and thus constitute the majority of flavonoids [75]. Here, we describe several flavonoids, including baicalin, quercetin, naringenin, silybin, genistein, and chrysanthemin, because of their relatively comprehensive research base and strong bioactivity (Table 2).

### 2.1. Baicalin

Baicalin is a flavone glycoside that can be hydrolyzed to baicalein [122]. Baicalin is widely distributed in the genus *Scutellaria* and is the main component of the fruit, root bark, and leaves of *Scutellaria baicalensis* [123]. As a glycosidic flavone, baicalin is hard to traverse the lipid bilayer by passive diffusion; hence, it is difficult to be absorbed in the intestinal tract [124]. Furthermore, pharmacokinetic studies have demonstrated that the absorption and elimination of baicalin are changed under different pathological conditions [125,126], suggesting that drug safety and possible individualized treatment requirements should be considered in the clinic.

The pharmacological effects of baicalin on atherogenic dyslipidemia and NAFLD have been reported in several studies. In 3T3-L1 pre-adipocytes, baicalin decreased the transcription of lipogenic genes CCAAT-enhancer binding protein α (C/EBPα) and peroxisome proliferator-activated receptor γ (PPARγ) [76]. Guo et al. showed that baicalin administration in high-fat diet (HFD)-fed rats relieved metabolic disorders and hepatic steatosis by enhancing adenosine monophosphate activated protein kinase (AMPK) phosphorylation and downregulating the expression of sterol regulatory element binding transcription factor 1c (SREBP-1c) and its target enzymes acetyl-CoA carboxylase alpha (ACACA) and FASN (fatty acid synthase) [77]. In another interesting study, baicalin also increased the expression of lipolytic factor peroxisome proliferator activated receptor α (PPARα) and its target gene carnitine palmitoyltransferases-1 (CPT-1) in carotid artery, and ultimately alleviated dyslipidemia in rats given a HFD [78]. Meanwhile, this effect of baicalin might also be related to the inhibition of the mitogen-activated protein kinase (MAPK) and nuclear factor kappa B (NF-κB) pathways [78]. Further exploration found that baicalin is also able to protect methionine–choline-deficient diet (MCD)-induced NAFLD by inhibiting the toll-like receptor 4 (TLR4) inflammatory signal in mouse liver [79]. Consistently, baicalin attenuates diet-induced NAFLD by suppressing the expression levels of pro-inflammatory cytochrome c oxidase subunit 2 (COX2) levels and pro-oxidative cytochrome P450 family 2 subfamily E member 1 (CYP2E1) levels, as well as the protein phosphorylation of c-Jun N-terminal kinase (JNK) in liver [80].

In accordance with the role of maintaining lipid homeostasis, baicalin can also improve diabetes and its complications. In human umbilical vein endothelial cells cultured with high glucose (HG), baicalin was shown to alleviate cellular oxidative stress by enhancing the nuclear factor erythroid 2 (NRF2)-mediated transcriptional activation of heme oxygenase 1 (HO-1), superoxide dismutase (SOD), and catalase (CAT) [81], which is expected to relieve aortic vascular injury in diabetic patients. Meanwhile, in differentiated C2C12 skeletal muscle cells, baicalin can also activate insulin receptor substrate 1 (IRS-1) and glucose transporter 4 (GLUT4), as well as AMPK, phosphatidylinositol 3-kinase (PI3K)/protein kinase B (AKT), and MAPK/extracellular signal-regulated kinase (ERK) signaling cascades, to enhance the glucose uptake and utilization of muscle cells [82]. Corresponding animal experiments also showed that baicalin has the potential to reverse HFD-induced hyperglycemia and systemic insulin resistance in mice, which might be attributed to the activation of the AKT/AKT substrate of 160 kD (AS160)/GLUT4 and MAPK/PPARγ coactivator 1α (PGC-1α)/GLUT4 pathways [83].

Although the data on the in vitro biological effects of baicalin are vast, up to now, very few clinical studies have been reported. In an earlier study, baicalin markedly reduced the serum levels of TG, total cholesterol (TC), and low-density lipoprotein (LDL)-cholesterol (LDL-C), but not HDL-C and apolipoproteins (APOs), in patients with coronary artery disease and rheumatoid arthritis, along with the inflammatory biomarkers cardiotrophin-1 (CT-1) and high-sensitivity C-reactive protein (hs-CRP) [25]. Although baicalin shows lipid-lowering and anti-inflammatory effects in circulating systems, clinical trials with NAFLD or diabetic patients are needed to reveal the real benefits of baicalin on MetS.

### 2.2. Quercetin

Quercetin is a typical example of a flavonol, mostly present in the form of glycosides [127]. It is the most abundant flavonoid in the human diet, with the average person consuming 10–100 mg/day from a variety of foods [128]. It is widely found in many plants and foods, such as red wine, onions, green tea, apples, sea buckthorn, hawthorn, and buckwheat [129]. Adverse effects of quercetin supplementation have rarely been reported in numerous published human experiments, and the effects are mild in nature [130].

Quercetin has attracted a lot of attention in recent decades since its therapeutic potential as anti-obesity, anti-diabetic, and lipotropic agent. For example, quercetin inhibits adipose tissue macrophage infiltration and the release of pro-inflammatory factors such as interleukin (IL)-6 and monocyte chemotactic protein 1 (MCP-1), so as to resist HFD-induced adipose tissue hypertrophy [84,85]. Dong et al. also suggested that the effect may be related to the activation of the AMPK/silent information regulator 1 (SIRT1) pathway [85]. In addition, adiponectin plays an important role in glucose and lipid metabolism with antiatherogenic and anti-inflammatory properties, while quercetin can stimulate its secretion in a PPARγ-independent manner [86]. In the liver, quercetin increased PPARα expression, inhibited lipogenic genes such as PPARγ, SREBP-1c, cluster of differentiation 36 (CD36), and FASN, accompanied by the upregulated antioxidant factors such as CAT and glutathione peroxidase 1 (GPX-1) and reduced pro-oxidant enzyme CYP2E1 [87,88]. Moreover, quercetin reduced the expression of TLR4 (mediates aortic lesions in the aorta of atherosclerotic rats) in human peripheral blood mononuclear cells (PBMC), resulting in inactivation of TLR4–MAPK/NF-κB signals, and ultimately contributing to the improvement of atherosclerotic inflammation [89].

On the contrary, physiological doses of quercetin increase the phosphorylation of AMPK, IRS-1, and AS160 in human primary myotubes, while quercetin exhibits hypoglycemic effects by increasing the expression of glycogen synthase kinase 3β (GSK3β) to promote glycogen synthesis after insulin stimulation [90]. Meanwhile, quercetin also reduces vascular endothelial injury by inhibiting the formation of myeloperoxidase/HG-dependent hypochlorous acid in streptozotocin-induced diabetic mice [131]. In addition, CYP2E1 activates the liver damage in stress-induced type 1 diabetes (T1D) liver damage. More importantly, this enzyme can be inhibited by quercetin to normalize the pro-oxidation-antioxidant system and prevent liver damage [91].

Quercetin also has the potential to reduce blood pressure. For example, quercetin weakens the hypertension induced by uterine perfusion pressure in pregnant rats by reducing the levels of endothelin 1 (ET-1) and endothelin receptor type A (ET-A) [92]. In addition, quercetin reduces the Na^+^ reabsorption in human renal epithelial cells and drops volume-dependent elevated blood pressure, and the underlying mechanism involved is activating Na^+^-K^+^-2Cl^−^ cotransporter 1 (NKCC1) and diminishing epithelial Na^+^ channel (ENaC) expression [93].

In humans, the intake of quercetin also shows multiple benefits. Oral administration with 150 mg/day quercetin for six weeks decreases systolic blood pressure (SBP) and plasma oxidized LDL (ox-LDL) concentrations [26], and in women with T2D, oral 500 mg/day quercetin significantly reduced the serum concentration of tumor necrosis factor α (TNF-α) and IL-6, as well as SBP [27]. Even at low doses of 60 mg/day, quercetin still reduced visceral fat and serum alanine transaminase (ALT) [28]. However, Verena Brüll et al. reported a test of ineffective quercetin administration in overweight-to-obese patients with pre- and stage I hypertension [29], which may be related to the shorter treatment cycle and lower dosage. Taken together, these results demonstrate the great potential of quercetin in the treatment of MetS, but efficacy needs to be further evaluated.

### 2.3. Naringenin

Naringenin, a dihydroflavanone, is a colorless and tasteless flavanone found mostly in the peels of citrus fruits and tomatoes [132]. In nature, naringenin comes in two forms: nonglycosylated (naringenin) and glycosylated (naringin or naringenin-7-O-glucoside) [132]. Only a handful studies on the safety, teratogenicity, and toxicity of naringenin have been issued, so its safety remains to be verified and caution should be exercised in the clinical use of naringenin [133].

Previously, naringenin was stated to enhance adiponectin transcription in differentiated 3T3-L1 cells [94]. Further studies confirmed that naringenin may increase the expression of hepatic PPARα and PGC-1α in rats fed a high-sucrose diet. As a result, the transcriptions of downstream fatty acid oxidation genes such as CPT-1, uncoupling protein 1 (UCP1), uncoupling protein 2 (UCP2), apolipoprotein A-I (APOA-I), and acyl-CoA oxidase (ACOX) were promoted, with the ultimate effect of reducing TG and TC content in plasma and liver [95,96]. Further research has revealed that the effect of naringenin on improving lipid metabolism is also attributed to the expressive suppression of liver X receptor α (LXRα) and its downstream lipogenic genes such as SREBP-1c, FASN, ATP-binding cassette sub-family A member 1 (ABCA1), ATP-binding cassette sub-family G member 1 (ABCG1), and 3-hydroxy-3-methylglutaryl-CoA (HMG-CoA) reductase (HMGCR) [96,97]. Moreover, naringenin alleviates liver peroxidation and inflammation via the inhibition of NF-κB signaling transduction in rats [98].

The inhibitory effect on atherosclerosis progression was also observed under naringin treatment. Naringin treatment may decrease the activity of acetyl-CoA acetyltransferase (ACAT), which mediates the formation of cholesterol ester, and downregulates the expression of the pro-inflammatory cytokines MCP-1 and vascular cell adhesion molecule 1 (VCAM-1) [99]. In addition, the inhibition of the inflammatory response and vascular smooth muscle cell (VSMC) proliferation might delay the development of atherosclerosis [134]. Our studies showed that naringenin enhances HO-1 expression and activity to inhibit TNF-α-induced VSMC proliferation and migration [100]. Additionally, naringenin is able to induce HO-1 expression by activating NRF2, thereby inhibiting the recruitment of white blood cells and reducing reactive oxygen species (ROS) production [101]. Accordingly, naringenin reduces the production of pro-inflammatory factors such as IL-33, TNF-α, IL-1β, and IL-6 by the inactivation of the NF-κB signaling pathway in macrophages [102].

In a double-blind cross-over study, hypertensives who received juice with different contents of naringin for five weeks showed lower SBP and diastolic blood pressure (DBP), and DBP was more effectively reduced in the high-dose naringin group [31]. Additionally, a clinical survey showed that daily supplementation with naringin led to a significantly marked antioxidant effect. For example, in the hypercholesterolemic subjects, dietary naringin effectively enhanced erythrocyte antioxidant enzyme activities and decreased serum TC and LDL-C, as well as apolipoprotein B (APOB) [30]. However, in another intervention of patients with hypercholesterolemia, no improvement of serum lipid homeostasis was found [32]. To date, only a few clinical researches are currently published, and further exploration is needed to achieve a viable and safe clinical treatment based on naringenin.

### 2.4. Silybin

Silybin, also known as silibinin, is the main representative of dihydroflavanols. The compound is composed of a mixture of two diastereomers in nature, Silybin A and Silybin B, with a molar ratio close to 1:1 [135]. Silybin was first extracted from the seed capsule of the plant *Silybum marianum* and is the major active constituent of silymarin [136]. Silymarin is safe in humans at therapeutic doses and is well tolerated even at a high dose of 700 mg daily for 24 weeks [137]. The same phenomenon occurred at 600 mg a day for 12 months [138]. Consequently, silymarin is considered to be a safe and well tolerated drug [139].

Although no evidence of an interaction with key pro-oxidant enzyme CYP2E1 has been found [140], silybin is commonly used in the treatment of fatty liver due to its lipolytic and anti-inflammatory properties. Our results showed that silybin is capable of enhancing the function of the LDL receptor (LDLR) by reducing proprotein convertase subtilisin/kexin type 9 (PCSK9) promoter activity, thereby limiting lipid accumulation in human hepatoma HepG2 cells [103]. Meanwhile, silybin confers resistance to hepatic steatosis, dyslipidemia, and inflammatory cell infiltration in vivo [104,141], which might be attributed to the reduced expression of TNF-α and IL-1β and the enhanced expression of anti-inflammatory factors IL-10 and adiponectin in adipose tissue [104]. Additionally, silybin enhances hepatic TG breakdown by modulating the expression of adipose TG lipase (ATGL) in rats with NAFLD [105]. More importantly, silybin treatment even significantly lowers the levels of *Firmicutes* and the ratio of *Firmicutes* to *Bacteroidetes* in the intestinal microflora, correcting the metabolic disturbance induced by HFD [142].

Furthermore, silybin has been proven to restore serum glucose, insulin, and glycosylated hemoglobin (HbA1c) in diabetic rats, along with reducing liver glucose output [143]. The equivalent improvement was observed in NAFLD rats, which might be related to the decreased expression of liver forkhead box O1 (FOXO1) and its target genes such as phosphoenolpyruvate carboxykinase (PEPCK), glucose-6-phosphatase (G6Pase), and other gluconeogenic genes [105]. Moreover, Xu and his colleagues even showed that this effect might be related to the expression of glucagon-like peptide 1 receptor (GLP1R) in the duodenum and subsequent neuronal activation in the solitary tract nucleus [106]. In contrast, Cun et al. showed that silybin is able to activate the NRF2-mediated antioxidant pathway to maintain the quality and function of pancreatic β-cells [107].

Two randomized clinical trials showed that silymarin, a complex mixture composed chiefly of silybin, might alleviate liver fibrosis based on histology, liver stiffness measurements, and the serum concentrations of hepatic enzymes [33,34]. Meanwhile, in a clinical observation performed by Chan and his colleagues, the administration of silybin with vitamins D and E for six months significantly improved metabolic markers, oxidative stress, and endothelial dysfunction in both NAFLD and MetS patients [35]. Nevertheless, considering that there is little clinical research on silybin monomer at present, silybin may be more used as a component of MetS drugs in the future.

### 2.5. Genistein

Genistein is one of the most abundant isoflavones in soybeans. Several studies have shown that genistein interacts with estrogen receptor in both animals and humans in a way that is similar to the effects of estrogen [144,145]. The intake of genistein in animals may affect the disruption of hormonal balance [146]. In addition, consumption of soybean as a supplement of genistein may cause minor stomach and intestinal side effects, and lead to allergic reactions [147]. Overall, more attention should be paid to the clinical use of genistein.

In HFD-fed mice, dietary intake of genistein reduced body weight (BW) and liver fat weight, as well as plasma and liver pyruvic aldehyde levels [108]. Mechanistically, genistein decreased the content of pyruvic aldehyde by the upregulation of glyoxalase 1 (GLO1), glyoxalase 2 (GLO2), and aldose reductase (AR), thus reducing the accumulation of pyruvic aldehyde-induced advanced glycation end products [108]. Choi and his colleagues showed that genistein also significantly inhibited lipid droplet formation in 3T3-L1 pre-adipocytes in a dose-dependent manner by reducing the expression of adipocyte-specific proteins such as PPARγ, C/EBPα, and fatty acid binding protein 4 (FABP4) and the lipogenic enzymes ATP citrate lyase (ACL), ACACA, and FASN [109]. In addition, a similar phenomenon was happened in NASH rats [110]. Genistein administration alleviated hepatic steatosis and apoptosis by downregulating PPARγ and upregulating adiponectin expression [110]. Gan et al. even showed that genistein promoted lipolysis in adipose tissue through miR-222, which targets an adipocyte proliferation-related gene BTG anti-proliferation factor 2 (BTG2) and a lipolytic gene adiponectin receptor 1 (ADIPOR1) [111].

Studies have shown that genistein promotes insulin secretion [148,149]. As early as 1993, Ohno et al. demonstrated that genistein can stimulate insulin secretion in the pancreatic β-cell line MIN6 through, at least in part, phosphodiesterase inhibition [112]. Subsequently, Liu et al. showed that genistein increases cellular cAMP accumulation and insulin secretion in the dose range of 0.01–5 μM, and this effect is due to the activation of protein kinase A (PKA) signaling [113]. In addition, genistein can regulate the composition of the gut microbiota. For example, genistein treatment reduces the ratio of *Firmicutes* to *Bacteroidetes* and the relative abundance of *Proteus*, thus improving the inflammatory response and insulin resistance of T2D mice [150].

In a randomized controlled trial involving obese people with insulin resistance, genistein increased skeletal muscle insulin sensitivity by raising the relative abundance of *Verrucomicrobia* in the gut microbiota of obese people and activating the expression of AMPK in skeletal muscle [36]. Similarly, genistein supplementation for twelve weeks in T2D patients significantly reduced serum levels of glucose, HbA1C, TG, and malondialdehyde (MDA), which is a marker for oxidative stress. As a result, the total antioxidant capacity (TAC) was raised, an event that might be useful in the control of metabolic dysregulation and oxidative stress in these subjects [37]. A similar phenomenon has been observed in patients with NAFLD. Oral administration with genistein for eight weeks was able to reduce insulin resistance, MDA, TNF-α, IL-6, and TG, along with an improvement in body fat percentage and the waist-to-hip ratio [38]. These results suggest that genistein is promising as a candidate drug for ameliorating dyslipidemia and hyperlipidemia, and could be applied in the treatment of MetS.

### 2.6. Chrysanthemin

Chrysanthemin, also known as cyanidin-3-O-glucoside (C3G), is one of the most common anthocyanins. C3G mainly exists in black soybeans, common beans, cowpeas, peanuts, lentils, and other plants [151].

Previous studies have shown that C3G treatment inhibits palmitic acid (PA)-induced lipid accumulation in 3T3-L1 pre-adipocytes through the inhibition of PPARγ and NF-κB inflammatory signals [114]. Subsequently, C3G increases the expression of UCP1 protein and beige adipocyte markers Cbp/p300 interacting transactivator with Glu/Asp rich carboxy-terminal domain 1 (CITED1) and T-box transcription factor 1 (TBX1) in 3T3-L1 cells, thereby inducing the formation of a beige phenotype [115]. In HepG2 cells, C3G significantly inhibits adipogenesis, and this effect is related to the inactivation of ACACA phosphorylation and overexpression of CPT-1 induced by increased AMPK activity [117]. In line with cell experiments, animal experiments have also confirmed that C3G relieves visceral fat and liver fat accumulation in obese mice, and the underlying mechanism is found to be partly related to the activation of lipoprotein lipase (LPL) in plasma and skeletal muscle, and the inhibition of LPL in adipose tissue following the activation of AMPK phosphorylation [118]. In addition, C3G also reduces overnutrition-induced inflammatory infiltration in adipose tissue. This effect is not only associated with decreased levels of inflammatory adipocytokines TNF-α, IL-6, and MCP-1, but also with decreased JNK activity, increased AKT phosphorylation, and enhanced nuclear exclusion of FOXO1 [119]. In particular, C3G enhances the transcription of UCP1 in beige adipose tissue (BAT) and subcutaneous white adipose tissue (SWAT), accompanied by the transformation of SWAT to BAT [116].

PA reduces the sensitivity of hypertrophic adipocytes to insulin signaling through the phosphorylation of IRS-1, while C3G reverses the adverse effects of PA, accompanied by a reduction in inflammatory infiltration through restoring IRS-1/PI3K/AKT activity [114]. Meanwhile, C3G can also improve systemic metabolism in *db/db* diabetic mice, regarding BW loss and the hyperglycemic effect, and increase glutathione (GSH) levels, improving diabetic nephropathy (DN) [152]. The same was observed in KK-Ay diabetic mice, where C3G improved hyperglycemia and insulin sensitivity by upregulating the expression of GLUT4 and downregulating the expression of retinol binding protein 4 (RBP4) and inflammatory indices (e.g., MCP-1 and TNF-α) in adipose tissue [120]. In addition, C3G had the power to ameliorate diabetes-related colonic dysfunction. Mechanistically, C3G increased the acetylation of FOXO1 through activating SIRT1 signaling, which in turn induced the expression and secretion of adiponectin, ultimately leading to the increase in the endothelial nitric oxide synthase (eNOS) expression and NO content [121]. In short, C3G shows a beneficial role in regulating the body’s energy balance and systemic metabolism. However, the lack of clinical studies has limited its further transformation and application.

## 3. Non-Flavonoids

Non-flavonoids mainly refer to a class of compounds containing one or more phenol groups but without the C6–C3–C6 structure. They are varied in category and structure, which ranges from simple to complex (Table 3). Non-flavonoids mainly consist of stilbenes, phenolic acids, and tannins, of which tannins can be further divided into gallotannin, ellagitannin, hydrolyzed and condensed tannin, and so on [153]. The representatives of non-flavonoids are phenolic acids, which are rarely found in a free form and are usually combined with other polyphenols, glucose, quininic acid, or structural components of the original plant [154]. Phenolic acids usually have two different parent skeletons: hydroxycinnamic acid and hydroxybenzoic acid [19,155]. Among them, gallic acid is considered the most important phenolic acid because it is the precursor of all hydrolysable tannins [156].

### 3.1. Resveratrol

Resveratrol is a stilbene compound found in plants such as peanuts, grapes, asters, and mulberries [208]. In plants, resveratrol has cis and trans isomers, namely, cis-resveratrol and trans-resveratrol, respectively [209]. However, resveratrol under natural conditions is mainly in the form of the trans isomer, the more biologically active isomer [210,211]. Therefore, resveratrol generally refers to trans-resveratrol. Clinical investigations showed that resveratrol supplementation exerts undesirable side effects. For example, the headache, dizziness, nausea, diarrhea, and abdominal discomfort, skin rashes, and even the elevation of liver enzymes [212,213,214]. Since these symptoms usually appear at high doses or short dosing interval, the side effects of resveratrol are mild and sporadic compared with its overwhelming health benefits.

In the past few years, resveratrol intervention has been demonstrated to effectively alleviate a variety of metabolic diseases. For instance, resveratrol significantly improves HFD-induced hepatic steatosis, as well as the accumulation of intestinal fatty acids and monoglycerides [157,215]. Mechanically, resveratrol is competent at activating the PKA/AMPK/PPARα signaling pathway in the liver, thereby restoring the activity of the mitochondrial respiratory chain against NAFLD in rats [157]. In hypercholesterolemia rabbits, resveratrol showed strong therapeutic potential by increasing serum levels of adiponectin and decreasing serum levels of leptin and insulin [158]. Notably, the intestinal flora is also an important target for resveratrol. Briefly, resveratrol treatment alters the composition of the intestinal microbiota, especially increasing the proportions of *Olsenella* and *Allobaculum*, which is useful for improving NAFLD and fasting glucose [216].

At the same time, resveratrol can also be used to treat diabetes and its complications. For example, it inhibits islet macrophage infiltration and β-cell death in T1D mice by inactivation of the C-X-C motif chemokine ligand 16 (CXCL16)/ox-LDL pathway or the CXCL16/NF-κB signaling pathway [159,160]. Not only that, but resveratrol also confers neuroprotective effects in diabetes-induced peripheral neuropathy by increasing NRF2 expression [161]. For the skeletal muscle system, resveratrol treatment restores the expression of the insulin receptor (IR) in the skeletal muscle of diabetic rats, whereas increases the expression of GLUT4 and tether containing a UBX domain for GLUT4 (TUG) in adipose tissue [162], which are beneficial for alleviating the metabolic disorders associated with diabetes to a certain extent. In *db/db* mice, resveratrol also increased serum adiponectin levels, accompanied by the high expression levels of ADIPOR1 and ADIPOR2 in the kidney to prevent the development of DN [163].

Resveratrol also has a hypotensive effect [217,218], and the mechanisms involved might be related to the levels of eNOS deacetylation [164]. In short, resveratrol plays a key role in activating the signals of AMPK, SIRT1, and NRF2, which induces eNOS de-acetylation and elevated NO production, ultimately resulting in vasodilation [165,166].

In a two-month pilot trial, resveratrol was certified to effectively decrease serum glucose and HbA1c levels. It also resulted in a decrease in the level of MDA and an increase in TAC in patients with T1D [39]. In patients with T2D, almost the same effect was happened [40,41]. However, an invalid prescription has been reported in T2D patients, which is due to the complexity of human genetics and other factors [42]. In addition, the effects of resveratrol on NAFLD progression are more complex [43,44,45,46,47,48,49,50,51,52,53], which means that the application of resveratrol for conversion needs to be further evaluated.

### 3.2. Gallic Acid

Gallic acid, also known as trihydroxybenzoic acid, is a member of the hydroxybenzoic acids [219]. It is widely found in grapes, chestnuts, sumac, witch hazel, tea, oak, and other plants [220]. Various research has suggested that gallic acid broadly exists in the form of free acids or various esters [221]. According to toxicity studies, GA scarcely exhibited obvious toxicity or side effects in a variety of animal experiments, which further clinical validation of its safety in humans is needed [222,223].

Previous lab studies have shown that although gallic acid treatment enhances adipose differentiation, it similarly induces the expression of adiponectin [167]. Subsequently, Tanaka and co-workers indicated that oral gallic acid may result in mice resisting the BW gain and impaired hepatic homeostasis induced by HFD. The expression analysis of hepatic steatosis-related genes showed that gallic acid treatment inhibits the expression of the ACACA and FASN genes in the liver [168]. Meanwhile, gallic acid supplementation significantly increases the expressions of medium-chain acyl-CoA dehydrogenase (MCAD), sterol regulatory element binding transcription factor 2 (SREBP-2), and HMG-CoA synthase (HMGCS) in the liver to promote fatty acid oxidation and ketogenesis, and decreases the expression of the de novo lipogenesis gene stearoyl-CoA desaturase 1 (SCD1) [169]. A recent finding indicated that gallic acid treatment inhibits the expression of IL-6, inducible nitric oxide synthase (iNOS), COX2, adhesion G protein-coupled receptor E1 (F4/80), and SREBP-1c in adipose tissue, thereby reducing the adipose tissue hypertrophy and inflammation induced by HFD [170]. Moreover, gallic acid also increases the expression of SIRT1 in the BAT of obese mice and enhances the thermogenic effect of BAT [171].

Consistent with the improvement in lipid metabolism, diabetes and its complications are significantly alleviated with gallic acid treatment. To be specific, gallic acid reduces glomerular mesangial matrix dilation and fibrosis in diabetic kidneys by downregulating transforming growth factor β1 (TGFβ1) signaling [172]. Meanwhile, gallic acid significantly improves the endothelial dysfunction in diabetic rats, and this protective effect might be mediated by increasing the plasma levels of miR-24 and miR-126, which are involved in vascular repair and anti-inflammatory effects [173]. Gallic acid also significantly activates GLUT4 expression in the adipose tissue of diabetic rats, as well as the enhancement of PI3K/AKT signaling, and ultimately leads to elevated sensitivity of adipose tissue to insulin and glucose uptake [174,175,176]. Interestingly, gallic acid has even exerted an ability to enhance β-cell survival, possibly by inhibiting the activity of the apoptotic factors caspase 3 and caspase 9 [177].

In an intervention study with gallic acid, the plasma concentrations of ox-LDL and hs-CRP were reduced after consumption of gallic acid in T2D patients. Furthermore, a significant reduction in oxidized purines and pyrimidines was also observed. The explanation is that daily supplementation of gallic acid might prevent oxidative DNA damage and reduce the markers that reflect inflammation [54]. In addition, a drink rich in gallic acid was shown to notably reduce the postprandial insulin incremental area and insulin secretion index, while elevated the insulin sensitivity index [55]. In one clinical trial with approximately thirty-seven individuals with MetS, the consumption of an açaí-based beverage (containing gallic acid) for twelve weeks significantly decreased the plasma level of interferon-γ (IFN-γ) and the urinary level of 8-isoprostane [56]. In short, these findings provide strong support for the clinical treatment of metabolic diseases such as obesity, dyslipidemia, and inflammation response.

### 3.3. Caffeic Acid

Caffeic acid is the major phenolic acid of coffee [224], which belongs to hydroxycinnamic acids, and is widely distributed in blueberries, coffee beans, sweet potatoes, and other plants [225]. In rats and humans, chlorogenic acid is rapidly metabolized to caffeic or ferulic acids after coffee consumption [226,227].

In a study by Kang et al., caffeic acid was found to have the ability to reduce BW and liver lipid degeneration in mice given a HFD [178]. By extension, caffeic acid effectively enhanced AMPK phosphorylation and reduced the levels of SREBP-1c, FASN, and ACACA. Meanwhile, caffeic acid supplementation was accompanied by the decreased expression of HMGCR and ACAT and the increased expression of PPARα and CPT-1 in mice [179]. The effect of caffeic acid on the improvement of alcohol-induced liver and kidney damage and dyslipidemia is also effective [228,229,230]. In a piece of work implemented by Kim et al., caffeic acid even induced autophagy-related protein 7 (ATG7), which is closely related to endoplasmic reticulum oxidative stress, thereby reducing liver dysfunction in HFD-fed mice [180]. Furthermore, caffeic acid also blocked monocyte transport to the resistin-activated endothelium by interfering with IL-8 signaling and inactivating the TLR4-NF-κB signaling pathway, thus exerting its therapeutic potential in the prevention of obesity-associated atherosclerosis.

Further research has shown that caffeic acid also has strong potential in the treatment of diabetes and its complications. In human endothelial cells, caffeic acid significantly reduces the growth inhibition and dysfunction of the endothelial cells induced by the HG environment [182,183]. Mechanically, endothelial cell growth is mainly related to the decreased activity of the caspase family and the increased phosphorylation level of B cell leukemia/lymphoma 2 (BCL-2) [182]. Additionally, the recovery of endothelial function is principally through the inhibition of NF-κB nuclear transport and the declining level of inflammatory cytokine endothelial leukocyte adhesion molecule 1 (ELAM-1), as well as the upregulation of the NRF2 pathway [183]. In *db/db* diabetic mice, caffeic acid significantly reduces blood glucose and HbA1c levels, while increasing serum insulin, C-peptide, and leptin levels [184]. Specifically, in the liver, caffeic acid supplementation significantly upregulates the level of glucokinase (GCK) and downregulates the expression of G6Pase, PEPCK, and glucose transporter 2 (GLUT2), thereby leading to reduced glucose output and uptake. Conversely, in adipose tissue, GLUT4 is significantly increased after caffeic acid treatment, which in turn promotes glucose utilization. In addition, caffeic acid enhances the expression of SOD, CAT, and GPX-1 in the red blood cells and liver of mice, hence boosting the antioxidant capacity of diabetic mice. Surprisingly, caffeic acid has been certified to regulate autophagy channels by inhibiting miR-636, which is involved in the development of DN, ultimately leading to an improved renal histological structure in diabetic rats [185].

As we known, coffee consumption is associated with a reduced risk of T2D and the mechanisms underlying the association have included attenuation of subclinical inflammation and a reduction in oxidative stress [57]. Briefly, positive alterations were observed for the serum concentrations of IL-18, 8-isoprostane, and adiponectin. Despite increased serum concentrations of TC and APOA-I, the ratios of LDL-C to HDL-C and of APOB to APOA-I decreased markedly [57]. Additionally, coffee intake might have an antiatherogenic property by enhancing HDL-mediated cholesterol efflux from macrophages via the presence of phenolic acids in the blood [58]. All of these effects are unlikely to be attributable to a single compound present in coffee; nevertheless, caffeic acid is expected to play a substantial role. Accordingly, caffeic acid remains potentially attractive in the treatment of metabolic disorders.

### 3.4. 1,2,3,4,6-Penta-O-Galloyl-D-Glucopyranose

1,2,3,4,6-penta-O-galloyl-D-glucopyranose (PGG) is formed by the condensation of five gallic acid molecules with D-glucopyranose, which largely exists in Chinese gallnut and is also the main representative of gallotannin. PGG exists in two anomeric forms, α-PGG and β-PGG [231]. While β-PGG can be found in a wide variety of plants, α-PGG is rather rare in nature [232].

In vitro studies have shown that β-PGG is a natural insulin analogue that induces the phosphorylation of IR in 3T3-L1 adipocytes, activates PI3K/AKT signaling, and promotes the membrane translocation of GLUT4 [186]. Meanwhile, β-PGG also alleviates lipid accumulation in 3T3-L1 cells dose-dependently by inhibiting the expression levels of MAPKs, PPARγ, and C/EBPα, along with decreasing the expression levels of ACACA, FASN, and SCD1 gene [187]. In addition, β-PGG inhibits the expression of IL-6 and MCP-1 in mature 3T3-L1 cells induced by TNF-α, followed by improving the inflammatory response [187]. Animal studies have also revealed that β-PGG significantly improves hypertriglyceridemia, hyperglycemia, and NAFLD induced by HFD [188,189]. Mechanistically, β-PGG reverses HFD-induced alterations in lipid-associated genes such as CD36, ABCA1, ACACA, cluster of differentiation 11c (CD11c), HMGCR, and microsomal triglyceride transfer protein (MTTP) [188]. Moreover, β-PGG inhibits the activity of 11β-hydroxysteroid dehydrogenases (11β-HSD-1), which participates in the regulation of the levels of glucocorticoids in the body [189].

Interestingly, α-PGG has the same target as β-PGG, but is more bioactive in stimulating glucose transport [186,233]. According to a recent study, α-PGG also inhibits the differentiation of 3T3-L1 pre-adipocytes into mature adipocytes [190]. Specifically, α-PGG inhibits adipogenesis and adipocyte differentiation by inhibiting the expression of the adipogenic factors PPARγ, C/EBPα, and mammalian target of rapamycin (mTOR), increasing the expression of preadipocyte factor 1 (PREF-1), and inducing cyclin-dependent kinase inhibitor 1 (P21)-mediated G1 phase cell cycle arrest [190]. Correspondingly, α-PGG significantly inhibits HFD-induced adipose tissue hypertrophy in mice by oral administration [190]. These data suggest that PGG, especially α-PGG, is promising as an effective therapeutic agent for the control of the chronic inflammation associated with obesity and insulin resistance. However, there are almost no clinical studies on PGG at present, which requires further clinical studies to fully understand its pharmacokinetics and pharmacodynamics in the human body.

### 3.5. Punicalagin

Punicalagin, the most abundant ellagitannin extracted from the fruit husk of pomegranates [234], is a polymer of ellagic acid and D-glucopyranose. Because of the existence of glycosyl in punicalagin, punicalagin is water-soluble, easily absorbed, and bioactive [235]. In laboratory research, punicalagin has been demonstrated to be an effective inhibitor of carbonic anhydrase and α-amylase; the inhibitory activity of punicalagin against α-amylase is comparable to that of acarbose [59,236].

In PA-induced HepG2 hepatocytes, punicalagin is thought to reduce lipotoxicity by restoring the expression of autophagy-related genes such as Beclin-1 and autophagy-related protein (ATG5) [191]. Yan and his colleagues have also indicated that the hepatoprotective effect was also associated with ERK and NRF2 activation [192]. Punicalagin also mitigates lipid formation and inflammatory response in the adipocytes of obese mice fed with HFD. In addition to the activation of NRF2 signaling, punicalagin also effectively downregulates the expression of NF-κB and inflammation-related cytokines such as TNF-α, IL-1β, IL-6, MCP-1, and NLR family pyrin domain-containing 3 (NLRP3) [193]. For the heart and liver, punicalagin has been demonstrated to modulate oxidative stress and mitochondrial biogenesis through the activation of PGC-1α and the NRF2 cascade, preventing the HFD-induced obesity-associated accumulation of lipid droplets, as well as organ damage [194,195]. Moreover, punicalagin also increased the secretion of adiponectin and the expression of its liver receptor ADIPOR2 [197].

Except for regulating lipid metabolism, punicalagin plays a hypoglycemic role both in vivo and in vitro. In HepG2 cells cultured in an HG environment, punicalagin upregulates the glucose uptake level [237]. Meanwhile, it was recently reported that the effect of punicalagin on reducing diabetes-related islet, liver, and kidney injury in diabetic mice can also be achieved by stimulating PI3K/AKT signaling and inhibiting the high mobility group box 1 (HMGB1)/TLR4/NF-κB pathway, which is involved in gluconeogenesis and the inflammatory response [197]. In a study by An et al., punicalagin also improved the glomerular interstitial hyperplasia and glomerular hypertrophy of diabetic mice via downregulating the expression of the pro-inflammatory factors IL-1β and NLRP3, as well as the pyroptosis-related factors caspase 1 and gasdermin D (GSDMD) [198].

In an adult population with atherosclerosis, supplementation with hydroxytyrosol and punicalagin markedly improved the early atherosclerosis markers involved in the asymptomatic phase. Briefly, after a twenty-week drug combination, the prehypertension and hypertension subgroups exhibited decreased SBP and DBP, as well as serum ox-LDL [60]. Kerimi and coworkers also demonstrated that pomegranate polyphenol significantly improves the incremental area under the curve for bread-derived blood glucose and peak blood glucose [59]. However, more clinical efficacy evaluations of punicalagin monomer should be conducted before drawing conclusions regarding the efficacy of punicalagin on MetS.

### 3.6. Curcumin

Curcumin, a rare pigment with a diketone structure in plants [238], is also one of the most widely sold natural food pigments in the world. It is often extracted from the rhizomes of turmeric and *Curcuma longa* [239]. Curcumin is recognized as a safe substance [240]. Studies have shown that curcumin is safe and tolerant well in both animals and humans [241,242,243,244]. In addition, they are nonmutagenic and safe during pregnancy in animals [245].

Curcumin has been shown to regulate various aspects of lipid and cholesterol metabolism [246,247]. In HepG2 cells, curcumin lowers oleic acid-induced lipid accumulation by reducing the expressions of SREBP-1c and FASN and enhancing the phosphorylation of AMPK and PPARα [199]. Similarly, curcumin significantly reduces BW gain and inhibits fat deposition in the adipose tissue of HFD-fed mice. This effect might be related to increased levels of AMPK and CPT-1, as well as the downregulated expression of the lipid-forming genes PPARγ, C/EBPα, and glycerol-3-phosphate acyltransferase 1 (GPAT1) [200]. Dietary supplementation with curcumin in MCD-fed mice also alleviates the severity of steatohepatitis to a certain extent by inactivating NF-κB activity and reducing the expression of pro-inflammatory cytokines such as intercellular cell adhesion molecule 1 (ICAM-1), COX2, and MCP-1 [201,202]. Meanwhile, the inhibitory effect of curcumin on liver inflammation is also associated with the decreased expression of CD11b, which participates in leukocyte activation, and tissue inhibitor of metalloproteinases 1 (TIMP1), a matrix-degrading-enzyme inhibitor [203]. In a study performed by Li et al., curcumin treatment upregulated the expression of the key antioxidant genes SOD1 and SIRT1, thereby enhancing the antioxidant capacity of the liver [204]. Curcumin also alleviated the hypertriglyceridemia and hepatic steatosis induced by a high fructose diet in rats, and the protective mechanism was achieved by inhibiting a cascade signal of protein tyrosine phosphatase 1B (PTP1B) [205], which is related to insulin and leptin signaling deficiency, and ultimately elevated the expression of PPARα.

Curcumin not only improves liver lipid accumulation and inflammation by inhibiting PTP1B but also reduces serum insulin and leptin levels via PTP1B [205]. The potential mechanisms involve enhanced IR, IRS-1, and Janus kinase 2 (JAK2) phosphorylation and activation of AKT and ERK signals, accompanied by inhibiting effects of signal transducer and activator of transcription 3 (STAT3) and suppressor of cytokine signaling 3 (SOCS3), ultimately leading to enhanced insulin and leptin signaling [205]. In addition, curcumin administration reverses the hyperglycemia and damage of pancreatic islets in diabetic mice induced by streptozotocin. In detail, curcumin blunts pancreatic lipid peroxidation, upregulates the activities of antioxidant enzymes (e.g., SOD, CAT, and GPX-1), and reduces the serum levels of TNF-α and IL-1β [206]. More importantly, dietary curcumin supplementation is also effective in *ob/ob* diabetic mice, which might be related to the increased expression of adiponectin in adipose tissues and the decreased levels of NF-κB and inflammatory factors TNF-α, SOCS3, MCP-1, and C-C motif chemokine receptor 2 (CCR2) in the liver, along with the decrease in the serum MCP-1 level [207].

Recent investigations indicate that curcumin also has beneficial effects in people with obesity or NAFLD. Bank et al. proved that curcumin supplementation in overweight and obese adolescent girls improves inflammatory markers hs-CRP and IL-6, along with TAC [61]. Meanwhile, they also demonstrated that curcumin has a substantial effect on cardiovascular risk factors, such as body mass index (BMI), waist circumference, hip circumference, HDL levels, and TG/HDL ratio [62]. Another research group showed that the serum levels of homocysteine and HDL were improved by twelve weeks of curcumin intervention, which might promote favorable cardiovascular health in young obese men [63]. However, no improvements in endothelial function or blood pressure were occurred with curcumin supplementation [63], contrary to the findings of Bank et al., which might be related to different genetic backgrounds or genders. Furthermore, in NAFLD patients, almost all metabolic parameters were improved [64,65,66,248]. In general, curcumin exerts great efficacy in improving obesity and regulating cardiovascular homeostasis. As for the increasing incidence of MetS, curcumin is expected to become the dominant drug in the treatment of metabolic disorders.

## 4. Other Issues of Natural Polyphenols

### 4.1. Polyphenol and CYP2E1 and Adiponectin Signaling in NAFLD

NAFLD, as the main manifestation and precursor of MetS in the liver, has complicated pathogenesis [249]. A series of previous studies have shown that CYP2E1 may be involved in excessive fat accumulation and aggravated oxidative stress-induced inflammation, leading to various liver cell damage and death [250,251,252]. Although only part of the polyphenols (including baicalin, quercetin) in NAFLD and MetS progression in regulating the expression of CYP2E1 and activity [80,88,91], a number of polyphenols (resveratrol and curcumin) interact with CYP2E1 in other diseases (e.g., alcohol-induced liver damage, chemically-induced HCC) [253,254,255]. Accordingly, the role and molecular mechanism of flavonoids in CYP2E1 regulation still need to be further explored [252]. On the other hand, adiponectin is the most common adipokine known to be inversely associated with insulin resistance, lipid accumulation, inflammation, and NAFLD development [256]. Our study supports the point that most polyphenols can regulate serum adiponectin levels and adiponectin signal transduction, thereby effectively improving the inflammatory response and preventing the progression of NAFLD in animal [86,94,104,110,121,158,163,167,196]. Of note, polyphenols exhibited great beneficial effects in cell and animal models, and clinical trials are not satisfactory. For example, only a few polyphenol treatments altered serum adiponectin levels in NAFLD patients and healthy volunteers [45,57], which may be related to the difference between human and animal. Hence, more human intervention experiments are needed to evaluate the effects of polyphenol on adiponectin in the future.

### 4.2. Polyphenol and Clinical Application of NAFLD

Existing clinical evidence shows that natural polyphenols significantly improve NAFLD. For example, curcumin and silybin both improved NAFLD in humans at different doses without significant gender bias [33,34,35,64,65,66,248]. However, the effects of resveratrol on patients with NAFLD are controversial [43,44,45,46,47,48,49,50,51,52,53]. Among them, resveratrol had no therapeutic effect on NAFLD patients at a low dose (150 mg/day) [44]. In contrast, at a dose of 300~600 mg/day, most of the studies showed that resveratrol has a good therapeutic effect, including but not limited to the loss of BS, decreased serum hepatic enzymes and lipid content, improved inflammatory signals, and relieved hepatic lipid accumulation [45,46,47,48,50]. However, it does not change the redox state in vivo [49]. In addition, the effects of resveratrol were inconsistent at doses ranging from 1.5 to 3 g/day, which may be related to high doses of resveratrol, or due to sample size and gender distribution [51,52,53]. Interestingly, in men, resveratrol has little positive effect on fatty liver and even increases serum hepatic enzyme levels [52,53]. Considering that both NAFLD and MetS are sex-binary disorders, this suggests that the effects of resveratrol are likely to be gender-biased. However, in clinical statistics, most of polyphenols were generally beneficial for both men and women [32,34,35,64,65,66,248]. Therefore, it is too early to conclude that resveratrol has a gender bias, because we cannot rule out the randomness issue of current clinical samples being too small (*n* = 8 or 10), and large-scale human studies are needed to prove it.

### 4.3. Polyphenol and NAFLD-HCC Progression

Regarding HCC, as one of the adverse outcomes of NAFLD, natural polyphenols are also expected to play an important role in the transition from NAFLD to HCC. A large body of evidence has accumulated over the past few years to suggest that inflammation-driven processes, such as the production of cytokines and chemokines, and the production of reactive oxygen species, contribute to the development of HCC [257,258,259]. As antioxidants and anti-inflammatory agents, polyphenols can remove excess oxygen free radicals and downregulate the expression of pro-inflammatory factors in the body, so as to maintain normal cell complexity and signal transfusion in the body and avoid damage to DNA, proteins, and lipids [260,261]. For example, resveratrol, previously mentioned, supplemented with 500 mg of resveratrol daily for twelve weeks improved inflammatory biomarkers and levels of hepatocyte apoptosis, thereby mitigating NAFLD progression [46]. On the side, disruption of the epigenome’s structure is thought to be the first step in the development of cancer [261]. Diets rich in polyphenols can act as epigenetic modulators to regulate the structures required by the epigenome to prevent cancer [262]. For example, patients with NAFLD have high levels of homocysteine [263,264], which affects the metabolism of CYP450 through DNA methylation modification, leading to and promoting the occurrence of liver cancer [265]. Curcumin treatment is able to reduce the blood homocysteine content in NAFLD patients [63]. Consequently, it is expected to play an important role in the progress of NAFLD-HCC. Unfortunately, the lack of sufficient in vivo studies limits the efficacy assessments of natural polyphenols against NAFLD-HCC progression.

### 4.4. Polyphenol and Drug Agents

Natural polyphenols possess great antioxidant and anti-inflammatory activities, which broaden their pharmaceutical applications to treat MetS, a well-known disease marked by low-grade and chronic inflammation. Despite such beneficial effects and the advantages of low capital cost and a vast range of plant sources [266], reproducing the beneficial effects of polyphenols in a clinical setting is more challenging than drug agents, and this may be due to a number of reasons. First of all, the content of polyphenols from different plant sources varies greatly, which increases our selection of extraction schemes, but also increases the complexity and time cost of preparation process [267]. Secondly, even for the same plant source, different preparation methods and drug delivery pathways in vivo will result in different bioavailability and even different product outcomes, which further limits experimental reproducibility and clinical transformation [268,269]. Finally, different genetic backgrounds and lifestyles may also lead to different pharmaceutical effects, and the time of follow-up may also affect the efficacy of drugs to some extent [270]. To address these issues, biochemists need to establish relatively reliable plant sources and standardized extraction and analysis steps to fundamentally reduce systematic errors [271,272]. Meanwhile, clinical observers need sufficient animal studies and extended follow-up studies of human beings to investigate the pharmacokinetics and pharmacodynamics of polyphenols in vivo [273,274]. Of course, for people with different living backgrounds, it is necessary for data scientists to conduct hybrid parameter correction and multivariate regression analysis to evaluate drug effects [275].

## 5. Conclusions

As mentioned earlier, different doses of drugs have different effects, and resveratrol is a good example. Meanwhile, different preparation and administration methods also affect the absorption and bioavailability of drugs to a certain extent. Therefore, the doses and delivery methods of these natural polyphenols should be finely scheduled for preventing or alleviating MetS. Additionally, due to the differences between human and animal models, even the population and gender differences, further large-scale clinical trials and efficacy evaluations are needed to elucidate the effects of natural polyphenols on MetS patients. Taken together, given the potential drug values of natural polyphenols, vast screening for identifying functional natural polyphenols will accelerate the development of new MetS drugs, thus benefiting patients suffering from MetS.

## Figures and Tables

**Figure 1 ijms-22-06110-f001:**
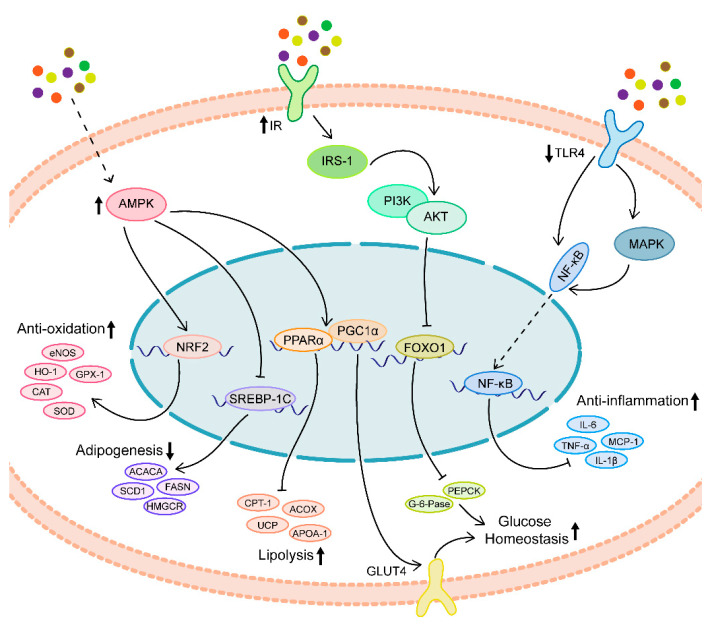
General overview of the underlying mechanism of natural polyphenols on MetS. Natural polyphenols improve the MetS mainly by increasing AMPK and IR/IRS1 activity and inactivating TLR4 pathway, which further induce the enhanced antioxidant and anti-inflammatory capacity.

**Table 1 ijms-22-06110-t001:** Local clinical trials with natural polyphenols on MetS and its components.

Author (Year) [Reference]	Disease Type and Sample Profiles	Series and Method	Findings
Hang (2018) [25]	Coronary artery disease and rheumatoid arthritis with a male predominance (*n* = 166)	500 mg/day baicalin for 12 weeks	↓ Serum TC, TG, LDL-C, APOs, CT-1, hs-CRP
Egert (2009) [26]	Overweight or obese subjects with a male predominance (*n* = 93)	150 mg/day quercetin for 6 weeks	↓ Serum ox-LDL, SBP
Zahedi (2013) [27]	T2D, women (*n* = 72)	500 mg/day quercetin for 10 weeks	↓ Serum TNF-α, IL-6, SBP
Nishimura (2019) [28]	Healthy subjects with a female predominance (*n* = 70)	60 mg/day quercetin for 12 weeks	↓ Serum ALT ↓ Visceral fat
Brüll (2017) [29]	Overweight-to-obese patients with pre- and stage I hypertension, both men and women (*n* = 70)	162 mg/day quercetin for 6 weeks	No effects on serum hs-CRP, TNF-α, leptin, adiponectin, glucose, insulin
Jung (2003) [30]	Hypercholesteremia, unknown gender distribution (*n* = 60)	400 mg/day naringin for 8 weeks	↓ Serum TC, LDL-C, APOB ↑ SOD, CAT
Reshef (2005) [31]	Stage I hypertension with a male predominance (*n* = 12)	Sweetie juice with 677 or 166 mg/L of naringin, 0.5 L/day for 10 weeks	↓ SBP, DBP
Demonty (2010) [32]	Hypercholesteremia, both men and women (*n* = 64)	500 mg/day naringin for 4 weeks	No effects on serum TC, TG, LDL-C, HDL-C
Solhi (2014) [33]	NASH with a male predominance (*n* = 33)	210 mg/day silymarin for 8 weeks	↓ Serum ALT, AST
Kheong (2017) [34]	NASH, both men and women (*n* = 49)	700 mg/day silymarin for 48 weeks	↓ NAFLD activity score, NAFLD fibrosis score, mean aspartate aminotransferase to platelet ratio index
Federico (2019) [35]	NAFLD (some with MetS), both men and women (*n* = 60)	606 mg silybin-phospholipid complex, 20 mg vitamin D and 30 mg vitamin E daily for 6 months	↓ Serum ALT, γGT, insulin, TNF-α, hs-CRP, TGF-β, IL-18, MMP2, EGFR, IGF2, HMGB1, Endocan, thiobarbituric acid reactive substances, insulin resistance
Cruz (2020) [36]	Obesity, unknown gender distribution (*n* = 22)	50 mg/day genistein for 2 months	↓ Insulin resistance, metabolic endotoxemia
Braxas (2019) [37]	T2D, women (*n* = 28)	108 mg/day genistein for 12 weeks	↓ Serum glucose, HbA1C, TG, MDA ↑ TAC
Amanat (2018) [38]	NAFLD with a male predominance (*n* = 41)	250 mg/day genistein for 8 weeks	↓ Serum insulin, TG, MDA, TNF-α, IL-6 ↓ Waist-to-hip ratio, body fat percentage, insulin resistance
Movahed (2020) [39]	T1D with a male predominance (*n* = 13)	1 g/day resveratrol for 2 months	↓ Serum glucose, HbA1c, MDA ↑ TAC
Abdollahi (2019) [40]	T2D with a male predominance (*n* = 35)	1 g/day resveratrol for 8 weeks	↓ Serum glucose, insulin ↑ HDL-C
Hoseini (2019) [41]	T2D and coronary heart disease, unknown gender distribution (*n* = 28)	500 mg/day resveratrol for 4 weeks	↓ Serum glucose, MDA, insulin, insulin resistance ↑ HDL-C, TAC
Bo (2016) [42]	T2D with a male predominance (*n* = 65)	500 or 40 mg/day resveratrol for 6 months	No improvement of the metabolic pattern of T2D
Mendía (2019) [43]	Dyslipidemia with a female predominance (*n* = 31)	100 mg/day resveratrol for 2 months	↓ Serum TC, TG
Kantartzis (2018) [44]	NAFLD with a female predominance (*n* = 53)	150 mg/day resveratrol for 12 weeks	No effects on liver fat content or cardiometabolic risk parameters
Chen (2014) [45]	NAFLD with a male predominance (*n* = 30)	300 mg/day resveratrol for 3 months	↓ Serum ALT, AST, glucose, LDL-C, TC, insulin resistance, TNF-α, IL-18, FGF21 ↑ Adiponectin
Faghihzadeh (2014) [46]	NAFLD with a male predominance (*n* = 25)	500 mg/day resveratrol for 12 weeks	↓ Serum ALT, inflammatory cytokines, nuclear factor κB activity, IL-18 ↓ BW, BMI, waist circumference, hepatic steatosis grade
Faghihzadeh (2015) [47]	NAFLD with a male predominance (*n* = 25)	500 mg/day resveratrol for 12 weeks	↓ Serum ALT ↓ Hepatic steatosis grade
Asghari (2018) [48]	NAFLD with a male predominance (*n* = 30)	600 mg/day resveratrol for 12 weeks	↓ BW, BMI
Asghari (2018) [49]	NAFLD with a male predominance (*n* = 30)	600 mg/day resveratrol for 12 weeks	No modification of oxidative/anti-oxidative status
Farzin (2020) [50]	NAFLD, both men and women (*n* = 25)	600 mg/day resveratrol for 12 weeks	↓ BW, BMI, waist circumference
Heebøll (2016) [51]	NAFLD with a male predominance (*n* = 13)	1.5 g/day resveratrol for 6 months	No improvement of histological features ↓ liver lipid content, serum ALT
Poulsen (2018) [52]	NAFLD, men (*n* = 8)	1.5 g/day resveratrol for 6 months	No changes in body composition or liver fat content
Chachay (2014) [53]	NAFLD, men (*n* = 10)	3 g/day resveratrol for 8 weeks	↑ Serum ALT, AST No improvement of any features of NAFLD
Ferk (2018) [54]	T2D with a male predominance (*n* = 19)	15 mg/day gallic acid for 7 days	↓ Serum ox-LDL, hs-CRP ↑ Cellular DNA stability
Costabile (2019) [54]	Healthy individuals, men (*n* = 12)	a drink with 1.562 g gallic acid equivalents for 7 days	↓ Postprandial insulin incremental area, insulin secretion ↑insulin sensitivity
Hyemee Kim (2018) [55]	MetS with a female predominance (*n* = 37)	açaí-beverage containing 1.139 g/L gallic acid, 325 mL/day for 12 weeks	↓ Plasma IFN-γ, urinary 8-isoprostane
Kempf (2010) [56]	Habitual coffee drinkers with a female predominance (*n* = 47)	600 mL filtered coffee daily for 1 month, followed by 1.2 L filtered coffee daily for 1 month	↓ Serum IL-18, 8-isoprostane, LDL-C to HDL-C ratio, APOB to APOA-I ratio ↑ Adiponectin, TC, HDL-C, APOA-I
Kondo (2010) [57]	Healthy volunteers with a male predominance(*n* = 8)	350 mL of freshly prepared coffee once	↑ HDL-mediated cholesterol efflux in macrophages
Kerimi (2017) [58]	Healthy volunteers, unknown gender distribution(*n* = 16)	200 mL pure pomegranate juice enriched in punicalagin twice	↓ Incremental area under the curve for bread-derived blood glucose, peak blood glucose
Fernández (2019) [59]	Seemingly healthy adults with a female predominance (*n* = 33/34)	585 mg punicalagin and 29.7 mg hydroxytyrosol daily for 8 weeks	↓ Serum oxLDL, SBP, DBP ↑ Flow-mediated dilatation
Bank (2019) [60]	Overweight and obese adolescent girls (*n* = 30)	500 mg/day curcumin for 10 weeks	↓ Serum hs-CRP, IL-6 ↑ TAC
Bank (2019) [61]	Overweight and obese adolescent girls (*n* = 30)	500 mg/day curcumin for 10 weeks	↓ Serum HDL, TG to HDL ratio ↓ BMI, waist circumference, hip circumference
Campbell (2019) [62]	Obesity, men (*n* = 22)	500 mg/day curcumin for 12 weeks	↓ Serum homocysteine ↑ HDL
Rahmani (2016) [63]	NAFLD, both men and women (*n* = 40)	70 mg/day curcumin for 8 weeks	↓ Serum TC, TG, LDL-C, ALT, AST, HBA1c, glucose ↓ Liver fat content, BMI
Mirhafez (2021) [64]	NAFLDs, both men and women (*n* = 40)	250 mg/day curcumin phytosome for 8 week	↓ Serum AST ↓ Hepatic steatosis grade
Chashmniam (2019) [65]	NAFLD, both men and women (*n* = 25)	250 mg/day phospholipid curcumin for 8 weeks	↓ Serum 3-methyl-2-oxovaleric acid, 3-hydroxyisobutyrate, kynurenine, succinate, citrate, α-ketoglutarate, methylamine, trimethylamine, hippurate, indoxyl sulfate, chenodeoxycholic acid, taurocholic acid, lithocholic acid
Panahi (2017) [66]	NAFLD, both men and women (*n* = 44)	1 g/day curcumin phytosome for 8 weeks	↓ Serum ALT, AST ↓ BMI, waist circumference

**Table 2 ijms-22-06110-t002:** Structure and classification of representative flavonoids and their mechanisms of action targeting MetS.

Flavonoids	Structure	Classification	Target of Drug Action	References
Baicalin	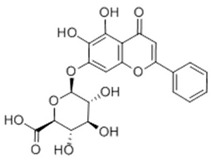	Flavone	↓ C/EBPα, PPARγ ↑ AMPK, ↓ SREBP-1c, ACACA, FASN ↑ PPARα, CPT-1, ↓ NF-κB, MAPK (carotid artery) ↓ TLR4 ↓ COX2, CYP2E1, JNK ↑ NRF2, HO-1, SOD, CAT ↑ IRS1, AMPK, GLUT4, PI3K/AKT, MAPK (myocyte)/ERK ↑ AKT/AS160/GLUT4, MAPK (muscle)/PGC-1α/GLUT4	[76] [77] [78] [79] [80] [81] [82] [83]
Quercetin	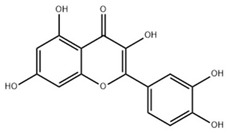	Flavonol	↓ IL-6, MCP-1, ↑ AMPK/SIRT1 ↑ Adiponectin ↑ PPARα, CAT, GPX-1, ↓ PPARγ, SREBP-1c, CD36, FASN, CYP2E1 ↓ TLR4-MAPK (PBMC)/NF-κB ↑ AMPK, IRS-1, AS160, GSK3β ↓ CYP2E1 ↓ ET-1, ET-A ↑ NKCC1, ↓ENaC	[84,85] [86] [87,88] [89] [90] [91]. [92] [93]
Naringenin	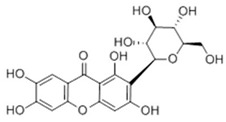	Dihydroflavanone	↑ Adiponectin ↑ PPARα, PGC-1α, CPT-1, UCP1, UCP2, APOA-I, ACOX ↓ LXRα, SREBP-1c, FASN, ABCA1, ABCG1, HMGCR ↓ NF-κB ↓ ACAT, MCP-1, VCAM-1 ↑ NRF2, HO-1 ↓ IL-33, TNF-α, IL-1β, IL-6	[94] [95,96] [96,97] [98] [99] [100,101] [102]
Silybin	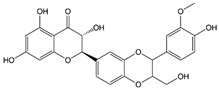	Dihydroflavanol	↓ PCSK9, ↑ LDLR ↑ Adiponectin, IL-10, ↓ TNF-α, IL-1β ↑ ATGL, ↓ FOXO1, PEPCK, G6Pase ↑ GLP1R ↑ NRF2	[103] [104]. [105] [106] [107]
Genistein	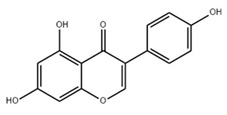	Isoflavone	↑ GLO1, GLO2, AR ↓ PPARγ, C/EBPα, FABP4, ACL, ACACA, FASN, SREBP-1c, ↑ AMPK ↑ Adiponectin, ↓ PPARγ ↑ miR-222, ↓ BTG2, ADIPOR1 ↓ phosphodiesterase ↑ PKA	[108] [109] [110] [111] [112] [113]
C3G	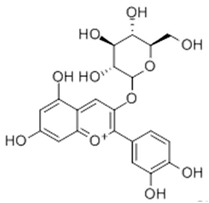	Anthocyanin	↓ PPARγ, NF-κB, ↑ IRS-1/PI3K/AKT ↑ UCP1, CITED1, TBX1 ↑ AMPK, CPT-1, ↓ ACACA ↑ AMPK, LPL (muscle), ↓ LPL (adipose tissue) ↓ TNF-α, IL-6, MCP-1, JNK, FOXO1, ↑ AKT ↑ RBP4, MCP-1, TNF-α, GLUT4 ↓ FOXO1, ↑ SIRT1, Adiponectin, eNOS	[114] [115,116] [117] [118] [119] [120] [121]

**Table 3 ijms-22-06110-t003:** Structure and classification of non-flavonoids and their mechanisms of action targeting MetS.

Non-Flavonoids	Structure	Structure	Target of Drug Action	References
Resveratrol	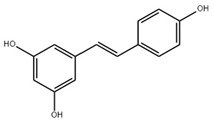	Stilbene	↑ PKA/AMPK/PPARα ↑ Adiponectin, ↓ Leptin, insulin ↓ CXCL16-ox-LDL/NF-κB ↑ NRF2 ↑ IR (muscle), TUG, GLUT4 (adipose tissue) ↑ Adiponectin, ADIPOR1, ADIPOR2 ↑ AMPK, SIRT1, NRF2, eNOS	[157] [158]. [159,160] [161] [162] [163] [164,165,166]
Gallic acid	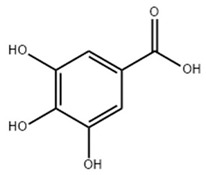	Hydroxybenzoic acid	↑ Adiponectin ↓ ACACA, FASN ↓ MCAD, SREBP-2, HMGCS, SCD1 ↓ IL-6, iNOS, COX2, F4/80, SREBP-1c ↑ SIRT1 ↓ TGFβ1 ↑ miR-24, miR-126 ↑ PI3K/AKT, GLUT4 ↓ Caspase 3, Caspase 9	[167] [168] [169] [170] [171] [172] [173] [174,175,176] [177]
Caffeic acid	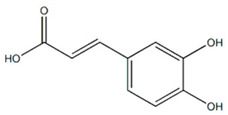	Hydroxycinnamic acid	↑ AMPK, ↓ SREBP-1c, ACACA, FASN ↓ HMGCR, ACAT, ↑ PPARα, CPT-1 ↑ ATG7 ↓ IL-8, TLR4-NF-κB ↓ Caspase family, ↑ BCL-2 ↓ NF-κB, ELAM-1, ↑ NRF2 ↑ GCK, SOD, CAT, GPX-1, ↓ G6Pase, PEPCK, GLUT2 (liver), ↑ GLUT4 (adipose tissue) ↑ miR-636	[178] [179] [180] [181] [182] [183] [184] [185]
PGG	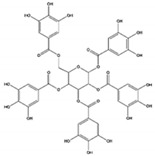	Gallotannin	↑ IR, PI3K/AKT, GLUT4 ↓ PPARγ, C/EBPα, ACACA, FASN, SCD1, IL-6, MCP-1, ↓ MAPK (adipocyte) ↓ CD36, ABCA1, ACACA, CD11c, HMGCR, MTTP ↓ 11β-HSD-1 ↓ PPARγ, C/EBPα, mTOR, ↑ PREF-1, P21	[186] [187] [188] [189] [190]
Punicalagin	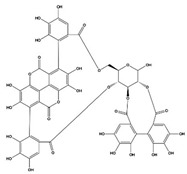	Ellagitannin	↑ Beclin-1, ATG5 ↑ NRF2, ERK ↑ NRF2, ↓ NF-κB, TNF-a, IL-1b, IL-6, MCP-1, NLRP3 liver ↑ PGC-1α, NRF2 ↑ Adiponectin (adipose tissue), ADIPOR2 (liver) ↑ PI3K/AKT, ↓ HMGB1/TLR4/NF-κB ↓ IL-1β, NLRP3, Caspase 1, GSDMD	[191] [192] [193] [194,195] [196] [197] [198]
Curcumin	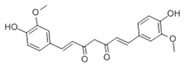	Diketone	↓ SREBP-1c, FASN, ↑ AMPK, PPARα ↑ AMPK, CPT-1, ↓ PPARγ, C/EBPα, GPAT1 ↓ NF-κB, ICAM-1, COX2, MCP-1 ↓ CD11b, TIMP1 ↑ SOD1, SIRT1 ↑ PTP1B, PPARα, IR, IRS-1, JAK2, AKT, ERK, ↓ STAT3, SOCS3 ↑ SOD, CAT, GPX-1, ↓ TNF-α, IL-1β ↓ NF-κB, TNF-α, SOCS3, MCP-1, CCR2 ↑ Adiponectin	[199] [200] [201,202] [203] [204] [205] [206] [207]

## Data Availability

Not applicable.

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
