# Peer review of "Natural Polyphenols in Metabolic Syndrome: Protective Mechanisms and Clinical Applications"

_ijms, 2021, doi:10.3390/ijms22116110_

Round 1

Reviewer 1 Report

GENERAL COMMENT

Natural polyphenols, biomolecules whose chemical structure is characterized by either single or multiple phenolic groups, can be classified into flavonoids and non-flavonoids. Natural polyphenols are widely found in cocoa beans, nuts, soybeans, olives, sesame seeds, tea, red wine, vegetables, and fruits. Based on their anti-inflammatory and antioxidant properties, natural polyphenols are awaited to protective from the MetS and its components.

This study reviews data regarding some flavonoids and non-flavonoids which are reportedly involved in the regulation of metabolism of glucose and lipids. The manuscript is timely and of interest given that a) the lay public feels strongly attracted by “natural” products as opposed to those synthesized in laboratory and b) the MetS, CVD and NAFLD have now reached epidemic proportions globally. The study has merit given that addresses a large body of literature and has more than 200 references. Among the weaker points of the study I report that it is not reader-friendly owing to unclear sub-heading and ambiguous sentences and should, therefore, be reworked to a quite large extent by putting some order in topics while guaranteeing accuracy in writing.

SPECIFIC COMENT

MAJOR

Lines 67-69 - This is a quite unconventional definition of the MetS that must be reworked based on standard notions (Circulation. 2009 Oct 20;120(16):1640-5). In doing so, please highlight the specificities of this syndrome in which individual components tend to anticipate the future occurrence of others (Dig Liver Dis. 2015 Mar;47(3):181-90). Additionally, please make it clear that cardiovascular diseases are not part of the syndrome although the MetS is deemed to be a risk factor for their development.

Line 69 “The incidence of MetS “à Is this a true incidence or a prevalence ?

Lines 66-96 – Describe here the notion that NAFLD has a mutual and bi-directional association with the MetS (reviewed in Int J Mol Sci. 2020 Aug 16;21(16):5888).

Given that this study reviews a substantial amount of information I suggest that using subtitles may greatly facilitate reading. For example analysis of each compound should have – a) Mechanism of Action – b) Studies on cell and animal models and – c) Clinical studies. Once this order is established in the text, it should be faithfully mirrored in title. For example it presently reads “Applications and pathophysiological mechanisms” while it should probably read “pathophysiological mechanisms and clinical applications”

Clinical studies, in their turn, should be classified as addressing MetS and its components as opposed to CVD (classified in primary and secondary prevention + treatment).

As regards NAFLD, I suggest to collect all clinical studies of polyphenols in human NAFLD in a Table. A new paragraph will comment on such a table [Include in this table – Author (year) Ref; Series and Method; Findings] though without duplicating data included it.

Both NAFLD and MetS are definite sexually dimorphic disorders (Lancet. 2020 Aug 22;396(10250):565-582). Is there any evidence that either sex is likely to benefit more from polyphenols ?

One of the most dreadful complications of NAFLD is hepatocellular carcinoma (HCC) which has a strong link with diabesity (Hepatoma Res 2020;6:83 DOI: 10.20517/2394-5079.2020.89 . Hepatology. 2016 Mar;63(3):827-38). Do these Authors believe that some polyphenols may play a role in chemoprevention of NAFLD-HCC ?

One of the main arguments in favor of natural substances contained in several foodstuffs would be their low cost as compared to drugs agents. Could these Authors address this topic ?

MIINOR

Throughout the manuscript make sure to re-write “nonalcoholic” rather than “non-alcoholic”: although both forms are commonly encountered, only the latter is consistent with pioneering definitions owing to Ludwig, Thaler and Shaffner (reviewed in Int J Mol Sci. 2020 Aug 16;21(16):5888).

“are expected to possess protective functions in the treatment of MetS.” Reword as follows: “are expected to be useful in the prevention and management of MetS”.

Metabolic syndrome (MetS) is a type of syndrome characterized by central obesity Please rephrase more concisely “Metabolic syndrome (MetS) is characterized by central obesity”

Unreferenced statements should either be deleted or supported by bibliography. For example “For individuals, MetS is associated with a wide range of physical and psychological health problems. Moreover, MetS puts enor mous pressure on the healthcare and social economy of the whole society. Therefore, it is an urgent and critical challenge for biologists to determine active pharmaceutical ingredients or pro-drugs that improve MetS and its complications.”

Figure 1 does not seem to a have an explanatory figure legend and it should.

“Natural polyphenols are a class of biomolecules widely found in plants, with more than 8000 species, most of which exist in cocoa beans, nuts, soybeans, olives, sesame seeds, tea, red wine, vegetables, and fruits” Of course wine is not a plant and, therefore this sentence should be better reworked.

Would suggest discussing the following studies: Biomed Pharmacother. 2020 Nov;131:110785. Biomed Pharmacother. 2021 May;137:111326.

Reviewer 2 Report

  1. From the title of the article it is not clear "pathophysiological mechanisms" refer to polyphenols or energy metabolism. If we are talking about natural polyphenols, then here we see a contradiction. The article deals exclusively with the protective functions of natural polyphenols (anti-inflammatory and antioxidant properties, regulation of glycolipid homeostasis). The participation of natural polyphenols in the induction and development of pathology is not considered.

If the pathophysiological mechanisms are related to energy metabolism (metabolic disorders, cardiovascular diseases), then this is a completely different topic. The article looked noticeably better if the authors defined the title of the article more clearly.

  1. The "keywords" section includes the term "metabolic syndrome". In the abstract, the interest in the metabolic syndrome is indicated. But the title of the article talks about energy metabolism. This is not quite correct. Energy metabolism is a broader term. Authors are encouraged to correct this. It will be more correct if the authors focus on the metabolic syndrome in the article.
  2. In the goal, the authors point out that the article is devoted to the consideration of possible mechanisms of action of natural polyphenols.

P 3 of 26

«In this context, we summarize some natural polyphenols (including flavonoids and non-flavonoids) that have been reported to be involved in the regulation of glucolipid metabolism and the improvement of CVDs, and review the possible mechanisms of each natural compound (Figure 1)».

There is no talk of a metabolic syndrome here. It may be worth pointing out the metabolic syndrome (in the abstract it appears), in which the effects and mechanisms of compounds are considered.

  1. The authors indicate only the potential medicinal value of natural polyphenols. Meanwhile, there is evidence from a safety study of natural polyphenols. There are quite a lot of published research results on this topic. Why do the authors avoid this aspect of natural polyphenol research? A small section on the biological safety of natural polyphenols is needed. This section will greatly increase the value of the article.
  2. The "Abbreviations" section is extremely inconvenient. Most of the abbreviations are clear. I suggest placing "Abbreviations" at the end of the article.

Round 2

Reviewer 1 Report

Submission is improved.